# Original Basic Activation for Enhancing Silica Particle Reactivity: Characterization by Liquid Phase Silanization and Silica-Rubber Nanocomposite Properties

**DOI:** 10.3390/polym14091676

**Published:** 2022-04-20

**Authors:** Enzo Moretto, Chuanyu Yan, Reiner Dieden, Pascal Steiner, Benoît Duez, Damien Lenoble, Jean-Sébastien Thomann

**Affiliations:** 1MRT Department, Luxembourg Institute of Science and Technology, 41 Rue du Brill, L-4422 Belvaux, Luxembourg; enzo.moretto@list.lu (E.M.); chuanyu.yan@list.lu (C.Y.); reiner.dieden@list.lu (R.D.); damien.lenoble@list.lu (D.L.); 2Goodyear S.A, Avenue Gordon Smith, L-7750 Colmar-Berg, Luxembourg; pascal_steiner@goodyear.com (P.S.); benoit_duez@goodyear.com (B.D.)

**Keywords:** silica, silica reactivity, silanization, organic-inorganic hybrids, silica-rubber composite

## Abstract

Silica fillers are used in various nanocomposites in combination with silanes as a reinforcing filler. In tire technology, silica is generally functionalized before (pre-treated) or during mixing (in-situ silanization or post-treated). In both cases, a soft base catalyst (e.g., triethylamine or diphenyl guanidine, DPG) is typically used to accelerate and increase the yield of the silane/silica coupling reaction. In this study, we investigated how pre-treatments of silica particles with either strong amine or hydride bases impact the silanization of silica prior to or during SBR mixing for silica-rubber nanocomposite fabrication. Our findings are supported by molecular characterization (solid state ^29^Si NMR, ^1^H NMR and TGA), and scanning electron microscopy. In addition, the impact of these silica pre-treatments on a nanocomposite’s mechanical properties was evaluated using dynamic mechanical analysis (DMA).

## 1. Introduction

Silane-modified silica particles are used in many different fields and applications, such as column chromatography [1], nanomedicine [2], concrete mix [3] or silica filler materials in tires [4]. The need for silanization is driven either by compatibilizing the filler with the matrix or by modifying the surface of the particles to create additional properties [5,6]. Silanization can be carried out in the gas, liquid or even solid phases when silica is mixed with the matrix at a high enough temperature. Chlorosilanes are widely used, but when mild reactivity, safer handling, and storage are required, alkoxysilanes like ethoxy or methoxysilane are preferred [7,8]. Silanization in liquid media is often carried out in aqueous or alcohol/water-based solutions or in an organic solvent [9,10,11], provided that the silane is soluble. Silanes are usually reacted via a hydrolysis-condensation reaction, and catalyzed either in acidic or basic conditions, hence the central role of water in the process. In dry organic solvent, chlorosilane does not react with hydroxyls unless water is present [10]. Alkoxysilane may react via the nucleophilic displacement of the alkoxy group, thereby generating alcohol [11], but the role of a catalyst appears essential, whether it is water or organic bases like amines. In the tire industry, the silanization of silica is usually catalyzed by mild amines, such as triethylamine or di-phenylguanidine and other works have investigated the impact of various amines on this process [12,13,14,15]. Increasing the silica reactivity is of interest for higher silane grafting efficiency. We hypothesize that the use of stronger bases could increase the amount and reactivity of silanol groups, and therefore silica. In this work, we investigated the impact of sodium hydride (NaH) and 1,5-diazabicyclo [5.4.0] undec-5-ene (DBU) as base catalysts to enhance silica reactivity. To our knowledge, these two base chemicals, and especially metal hydrides, have never before been used for silica reactivity enhancement. The impact of NaH and DBU on the morphology and surface of silica particles was examined with scanning electron microscope images and ^29^Si NMR. Then, to evaluate the effects of the bases on silica reactivity, we first used silica silanization as a simplified quantitative model. Silane grafting was quantified by ^1^H-NMR and TGA. Base catalyst-loaded silicas were then incorporated into rubber nanocomposites to create a more complex model to account for the effect of the catalysts. The effect of silica reactivity enhancement was evaluated via the mechanical properties of the materials obtained, using dynamic mechanical analysis (DMA).

## 2. Materials and Methods

Precipitated silica (10 nm primary particle size, 200 m^2^/g), 1,5-diazabicyclo [5.4.0] undec-5-ene (DBU), triethylamine (TEA), sodium hydride (NaH), tetrahydrofuran (THF), toluene, cesium fluoride, deuterated water (D_2_O), deuterated hydrochloric acid 35 wt. % (DCl), deuterated chloroform (CDCl_3_), and octamethylcyclotetrasiloxane (OMCTS) were purchased from Merck (Darmstadt, Germany). Hexyltrimethoxysilane (C6), dodecyltrimethoxysilane (C12), and octadecyltrimethoxysilane (C18) were purchased from Gelest (Morrisville, PA, USA).

For the fabrication of a silica-rubber composite, pre-made base-loaded silica, poly-styrene-butadiene (SBR) and polybutadiene rubber (BR), treated distillate aromatic extracted (TDAE) lubricating oil, zinc oxide, stearic acid, N-(1,3-dimethylbutyl)-N′-phenyl-p-phenylenediamine (6-PPD), sulfur, 2-mercaptobenzothaizole (MBT), di-phenylguanidine (DPG), and N-cyclohexyl-2-benzothiazolesulfenamide (CBS) were used.

### 2.1. Silica Pre-Loading of the Base Catalyst

Two organic bases were selected, 1,5-diazabicyclo [5.4.0] undec-5-ene (DBU) and triethylamine (TEA), as well as the inorganic base sodium hydride (NaH) base, to catalyze the silanization reaction of silica particles. We followed a silica pre-loading method, in which the particles were first blended with an organic base in a solvent, prior to the actual silanization. 

1 g of silica particles was suspended in 100 mL of THF and was stirred vigorously at room temperature. Then, 6 mmol of DBU, NaH or TEA were added, and the mixture was further stirred for 15 min. In the specific case of sodium hydride being used, gas bubbles were observed when silica and NaH were mixed together. The suspension was then dried under reduced pressure to remove all solvent. The pre-loaded base silica particles were stored under argon in a sealed container.

### 2.2. Silica Silanization

For the silanization reaction, 1 g of silica particles in a 250 mL round-bottom flask was mixed with 100 mL of toluene. The mix was brought to reflux and 0.78 mmol of either C6, C12 or C18 silane was added. The reaction was carried out at reflux temperature for 24 h under vigorous stirring. Then, the silanized silica was separated by centrifugation at 10,000× *g* for 3 min, the liquid was poured away, and fresh toluene was added. The centrifuge tube was shaken vigorously, and this washing cycle is repeated twice. Subsequently, the moist silanized silica particles were dried under reduced pressure and stored in a glass vial under inert argon atmosphere.

### 2.3. Preparation of ^1^H NMR Samples

First, simple silane solutions were prepared for the identification of the silane’s signals. 0.01 mmol of each silane was solubilized in 1 mL of deuterated chloroform (CDCl_3_). 

To obtain the spectrum of the silane after the reaction with the cesium fluoride/deuterated hydrochloric acid (CsF/DCl) solution, silane was reacted with the CsF/DCl solution. The subsequent solubilized silane residue signal obtained was compared to the internal standard signal by reacting 30 mg of silane with 1 mL of a 310 mg/mL CsF solution in deuterated water (D_2_O) and 260 µL of DCl (35 wt. %) in a 2 mL PP tube. The tube was left to react for 24 h at room temperature. Subsequently, 1 mL of 0.01 mmol/mL octamethylcyclotetrasiloxane in deuterated chloroform (OMCTS in CDCl_3_) solution was added to the tube. The tube was shaken vigorously by hand and left to settle for the two phases to separate. A 600 µL sample of the chloroform phase was poured into a glass NMR tube suitable for ^1^H NMR analysis.

For silanized-silica grafting quantification, 10 to 20 mg of silanized silica were added into a 2 mL PP-tube. Then, 1 mL of a 310 mg/mL CsF solution in D_2_O and 260 µL of DCl were added. The solution was left to react under stirring for 24 h at room temperature. Subsequently, 1 mL of an OMCTS-CDCl_3_ solution at 0.01 mmol/mL was added, the tube was shaken vigorously and a 600 µL sample of the chloroform phase was poured into a glass tube suitable for ^1^H-NMR.

### 2.4. Silica-Rubber Composite Fabrication

All ingredients were mixed using a HAAKE™ PolyLab™ QC ThermoScientific (ThermoScientific, Waltham, MA, USA) internal mixer and a roll mill at each mixing step. Component quantities and mixing steps were summed up in Table 1 below. Quantities are given in phr, a mass unit meaning per hundred rubber. Thus, 80 phr of silica means that for a total of 100 g of rubber, 80 g of silica are used. All green composite materials were cured in a hydraulic press at 170 °C for 10 min. at 150 kPa.

### 2.5. Characterization

The mass loss of silanized silica powders was recorded on a Mettler Toledo TGA 2 (Mettler-Toledo, Columbus, OH, USA) in an alumina crucible, at a heating rate of 10 °C/min from 25 °C to 1000 °C, with a 20 min stabilization step at 25 °C and under nitrogen atmosphere. 

The mechanical properties of the resulting composite materials were tested by dynamic mechanical analysis (DMA) at 1 Hz and 10 Hz, at a free length of 5 mm and a temperature sweep from −80 °C up to 100 °C. 

NMR spectroscopy was performed on a Bruker AV III 600HD instrument (Bruker, Billerica, MA, USA) (600 MHz ^1^H frequency). ^1^H NMR spectra were recorded using a 5 mm BBO probe with z-gradients at 298 K using a 30° pulse, a sweep width of 40 ppm, and 8 transients (160 k datapoints each) were acquired. The data were processed using Topspin 3.5pl7, which zero filled once and multiplied with an exponential line-broadening function of 0.50 Hz prior to Fourier-transform. 

^29^Si MAS NMR spectra were recorded on the same spectrometer with a 4 mm H/X probe, using direct polarization (HPDEC pulse program) at 303 K and 8 kHz MAS frequency. The ^29^Si 90° pulse was 2.5 µs, and a decoupling field of 100 kHz was applied during the acquisition of 900 transients (each with 2048 datapoints), separated by a 120 s relaxation delay and a sweep width of 300 ppm centered on −50 ppm.

## 3. Results and Discussion

### 3.1. Impact of Basic Activation on Silica Particles

The SEM images (Figure 1) show non-treated silica, DBU, and NaH-loaded silica at two different magnifications. When base-loaded silicas are compared to the untreated silica, the particles look less spherical. We attribute this morphological change to the chemical interaction of the bases with the particles. Additionally, the SEM images of DBU-loaded silica show the presence of residual DBU molecules in the form of a white cloudy halo surrounding the silica; this is typical of organic matter under an electron beam and is not observed when sodium hydride is used.

^29^Si ssNMR confirms a chemical reaction between silica and sodium hydride NaH. The Q3 signal at −102.5 ppm (see Figure 2) clearly shows an increase in Si-OH content for NaH-loaded silica. However, DBU does not seem to have the same effect on silica. We initially hypothesized that both bases would react with the surface water of the silica and form HO^−^ hydroxide ions. Indeed, hydroxide solutions are capable of hydrolyzing amorphous silica [16], however, our conditions (stirring at room temperature for 15 min) do not fulfill the thermodynamic requirement for this reaction. Another possible reaction is the acido-basic interaction of DBU or NaH with silica. If the alkaline character of DBU or NaH was responsible for the extra silanol production, the rise of the Q3 signal would be observed in both cases. Despite both bases having a high enough pKa (13.5 for DBU [17,18] and 50 for H_2_ [19,20], the conjugated acid of the hydride H^−^) that are sufficient to deprotonate Si-OH silanols and form the conjugated base Si-O^−^, the increase in the Q3 signal is only observed in the case of NaH. Because of this, the production of silanols is not a result of the alkaline character of sodium hydride, but rather, of its nucleophilicity. On the other hand, DBU is a poor nucleophile since its basic nitrogen atom is obstructed by the structure of the molecule. Therefore, we suggest that a possible explanation for the action of sodium hydride on silica is that the nucleophilic H^−^ of sodium hydride reacts with the Si-O-Si siloxane bridges of the silica in a nucleophilic attack and generates Si-H and Si-O^−^Na^+^. Finally, Si-H reacts with HO^−^ or H_2_O to form Si-OH and H_2_ [21], which would explain the observed bubbling and the increase of the Q3 signal in the NMR spectra. The expected presence of Na^+^ or DBU^+^ counter ions at the silica surface would change the particle’s cohesion behavior, since less Si-OH are available for hydrogen bonding, thus reinforcing our observations on the morphology change of the silica particles discussed earlier and shown in Figure 1.

### 3.2. Silica Reactivity Enhancement Evaluation

We decided to evaluate the effect of bases on silica reactivity using two models. The first one quantifies the yield of a simple silica silanization in a liquid organic solvent. In the second model, we used the base-loaded silica for the fabrication of a silica-rubber composite and measured the mechanical properties of the corresponding materials.

Silanization generally designates the grafting of organosilanes on a substrate to modify its surface properties. The yield of this reaction is directly influenced by the number of reactive sites on silica (i.e., silanols) and their reactivity. Silica particles were reacted with simple alkyltrimethoxysilanes of different lengths in the organic solvent toluene. The main reason for choosing these conditions was to minimize the self-condensation of the silanes that could then react with silica. Indeed, the self-condensation of silanes generates silane oligomers that are still capable of reacting with silica particles. This phenomenon causes the grafting of more silane onto silica, irrespective of the silica reactivity, and artificially boosts the silanization yield. To quantify the silanization yield, we chose two characterization methods, i.e., thermogravimetric analysis (TGA) and ^1^H NMR.

### 3.3. Thermogravimetric Analysis

Performing thermogravimetric analysis on silanized silica at sufficiently high enough temperatures allows us to measure the mass loss associated with silane pyrolysis. However, the calculation of the silanization yield is not straightforward, as other phenomena can occur simultaneously during heating, like in our case, the desorption of volatile compounds and dehydroxylation of the silica itself. Although these different events happen at determined temperature ranges, they can overlap, as is well documented with the case of silica dehydroxylation [22]. Dehydroxylation occurs at temperatures ranging from 110 °C to 1000 °C and the amount of water generated depends on the type of silica. To account for the mass loss due to the dehydroxylation of the silica during the silane pyrolysis, we recorded the thermograms of the untreated silicas and subtracted them from those of the silanized silicas. The silane mass-loss is calculated, for all samples, by subtracting the residual mass at 600 °C from that at 400 °C. Grafting yields are then calculated by dividing the mass loss between 400 °C and 600 °C by the molar mass of the organic residual part of the silane molecule. This calculation method was discussed by Kunc et al. [23], and can be considered correct if we hypothesize that all silanized-silica samples undergo the same dehydroxylation under heating. However, as in the case of silanized silica, that this is not the case, there is also an argument against this method, as the silanization reaction consumes hydroxyl groups on the surface of the silica, and therefore lowers the final number of hydroxyl groups to be dehydroxylated. Also, the remaining unreacted alkoxy groups of silanes still present on the final material account for part of the recorded mass loss. This can lead to an over-estimation of the mass loss attributed to the grafted silane. Moreover, this method cannot be used in the case of multiple and different silanes being grafted onto silica particles as all silanes would decompose over the same temperature range, unless an additional characterization method is potentially used in parallel with TGA, such as infra-red or mass spectrometry. TGA thermograms for C18-silanized silica with different base catalysts are shown below in Figure 3.

From the analysis of the thermograms, we observe that the use of a base catalyst considerably increases the amount of grafted silane. The difference between NaH and DBU is less obvious due to the degradation of the remaining storage oil of NaH and DBU residue. Appendix A show TGA thermograms for C12 silanized silica activated with NaH and DBU, respectively. Plotting dTGA vs. temperature helps to better differentiate the degradation phenomenon occurring. In the case of NaH (Appendix A), from 25 to about 100 °C, the mass loss is attributed to moisture and the solvent remaining on the silica. The two peaks centered at 300 °C and 400 °C, respectively, are present for both NaH-activated silica and silanized silica. We attribute these two peaks to the remaining silicon oil used for the storage and conservation of NaH. Finally, the peak centered at 500 °C and only present for silanized silica is attributed to the degradation of the grafted silane. For DBU in Appendix A, the same peak from 25 to 100 °C is attributed to moisture and residual solvent. The peak at 150–240 °C is attributed to the remaining DBU molecules. It may be noted that with both bases, the base residue is lower for silanized silica. The counter ions DBU^+^ and Na^+^ are expected to be released after the SiO^−^ reaction with the silane. In addition, silanized silica has seen more washing steps, helping to remove oil residues for NaH. Finally, for Appendix A, the major mass loss at 300 °C, only present in the case of silanized silica, is attributed to the degradation of the silane.

### 3.4. ^1^H Liquid State NMR

Pure silanes were reacted with the CsF/D_2_O/DCl solution for the identification of peaks, as shown in Figure 4. Alkylsilane (C6, C12 and C18) peaks are identified as follows: 0.64 ppm Si-CH_2_, 0.88 ppm CH_3_, 1.2 to 1.45 ppm aliphatic CH_2_, 1.57 ppm residual water, and 3.56 ppm are O-CH_3_ groups. Octamethylcyclotetrasiloxane was chosen as the internal standard because of its good solubility in chloroform, and the fact that it gives rise to only one signal (0.09 ppm CH_3_ of OMCTS) in a range that does not overlap with the signal from the samples. 

We first observe that the peak at 3.56 ppm disappears after the hydrolysis of the methoxy groups of methoxysilanes, and we do not observe a peak at 3.49 ppm that could be attributed to methanol in solution. We suggest that the methanol remained in the aqueous phase during the extraction with chloroform because of its polarity. The peak at 0.64 ppm is also absent. We could not find an explanation for this, as we do not expect to break a C-C bond in these conditions. Unlike Yang et al. [24], we do not observe any signal being the sign of alcohol in solution that would be the results of the opening of the Si-C bond, and oxidation of the carbon into alcohol. Practically, when pure silanes or silanized silica were reacted with CsF/D_2_O/DCl after 24 h, a thin white hydrophobic film appears on the tube walls. For a control sample consisting of unmodified silica mixed with the CsF/D_2_O/DCl solution, no such film was not observed. Upon contact with the deuterated chloroform extraction solution, this film solubilizes and is absent from the tube walls. We identify this as a sign of the reaction of the silane with cesium fluoride and hydrochloric acid and their removal from the silica. We also observe that prior to the reaction, modified silica wettability toward the aqueous solution was very poor, as silica floated onto the liquid and did not get wet when vigorously shaking the tube, as is expected from hydrophobized silica. After the 24-h. reaction, silica particles displayed good wettability toward the aqueous solution and sank to the bottom of the tube, consolidating the idea of the silane removal from the silica. 

For the integration of the signals, the peak at 0.88 ppm was chosen to quantify the amount of silane in solution. Typical ^1^H NMR spectra of silanized silica with C6, C12, and C18 are shown in Figure 5. Peaks of interest are integrated, and numerical values are obtained in Equation (1) below.
(1)$$nsilane=\frac{ACH3}{AOMCTS}\times \frac{NOMCTS}{NCH3}\times nOMCTS$$
where
*nsilane* is the amount of silane in solution (mmol),*ACH*_3_ is the area of the CH_3_ peak at 0.88 ppm,*AOMCTS* is the area of the OMCTS peak at 0.09 ppm,*NOMCTS* is the number of hydrogen nuclei generating the signal,*NCH*_3_ is the number of hydrogen nuclei generating the signal,*nOMCTS* is the amount of OMCTS internal standard in the CDCl_3_ solution (mmol).

Quantitative results from TGA and ^1^H NMR are displayed in Table 2 and Figure 6. It must be noted that TGA values are always higher than the corresponding ^1^H NMR values. The reasons cited earlier, i.e., residual base catalyst, water from silica dehydroxylation, and unreacted alkoxy groups may account for a fraction of the measured weight loss but do not actually represent silane grafting. As for the silane length, the longer it is, the lower the grafting yield. This trend is to be expected due to the bulkiness of the long silane, which reduces the reaction kinetic via steric hindrance. 

Through the quantification of silanization on base-loaded silica we already have elements to comment on the respective effect of DBU and NaH. DBU maximizes grafting, followed by NaH, and finally TEA, as follows: no base < TEA < NaH < DBU.

### 3.5. Evaluation of Silica Reactivity Enhancement in Rubber-Silica Composites

The effect of base-loading for the in-situ silanization of silica was measured using dynamic mechanical analysis to test the nanocomposite’s mechanical properties, which reflect the way filler particles interact with the matrix. Indeed, the motion of polymer chains increases as the temperature increases. The glass transition temperature T_g_, defined in DMA by the position of the tan(δ) peak, is the temperature range where the polymer transitions from a glassy to visco-elastic state. The more polymer chains participating in the transition, the higher the T_g_ peak. It has been demonstrated that a strong polymer-filler interaction can prevent a fraction of the chains from moving [25,26], thus lowering the T_g_ peak in DMA. In silica-filled rubber composites, filler-polymer interactions originate from rubber trapped between filler aggregates and rubber bonded to silica via the interaction of the silane with the fillers and the polymer, which increases the storage modulus of the composite [26,27,28,29].

Of the three samples tested, NaH-loaded silica demonstrated the highest conservation and loss module, followed by the untreated silica, and finally DBU-loaded silica. When looking at tan(δ), the base-loading of silica clearly affects the visco-elastic behavior of the composite material. In our case, the base activation of silica increases the filler-polymer interactions, as T_g_ peaks in Figure 7 are lower for DBU and even lower for NaH-loaded silica. Interestingly, DBU is better for silane grafting in the liquid phase, but NaH provides better reinforcement properties to the composite. One hypothesis is that the DBU^+^ counter ion, because of its size, has more shielding power in a polymer blend than Na^+^ and thus prevents, to some extent, the reaction of silanes with the surface of silica in rubber. In a liquid phase silanization, in which the molecular mobility is much greater, this effect may be minimized. Another hypothesis is that it is easy for NaH-loaded silica, thanks to its extra silanols, to generate aggregates trapping the rubber matrix, thus resulting in this additional stiffening effect.

## 4. Conclusions

The present study investigated the effect of two unusual base catalysts, DBU and NaH, for silica silanization in the liquid phase, as well as in situ during the curing of SBR-silica nanocomposite. DBU demonstrates a higher grafting yield than NaH and TEA, and the size of silane influences the amount grafted onto the silica particles. This was confirmed by the ^1^H-NMR and TGA silanization yield quantification, despite the discrepancy still present between the two techniques. Base-loaded silica were included in the fabrication of nanocomposites, and their subsequent mechanical properties were measured by DMA to study the impact of base loading. Contrary to the trend observed in liquid phase silanization, NaH-loaded silica yielded better reinforcing properties than DBU. We hypothesize that the DBU counter ion can shield silane from silica in the polymer melt, and that extra silanols from NaH-loaded silica can generate more silica aggregates in the composite and trap more rubber, bringing a higher degree of reinforcement to the composite.

## Figures and Tables

**Figure 1 polymers-14-01676-f001:**
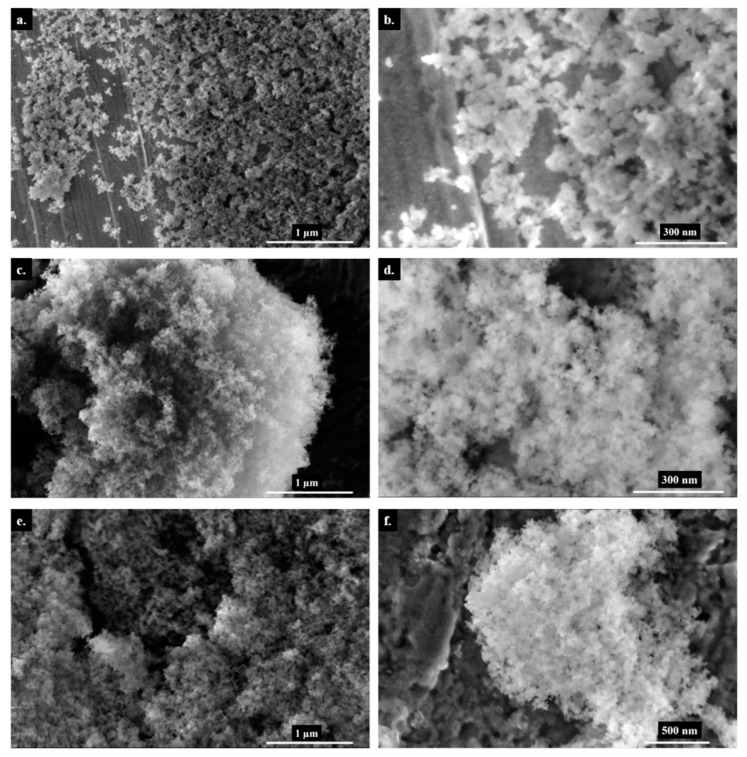
Scanning Electron Microscopy images of silica particles—Non-treated (**a**,**b**), DBU−loaded (**c**,**d**) and NaH−loaded (**e**,**f**).

**Figure 2 polymers-14-01676-f002:**
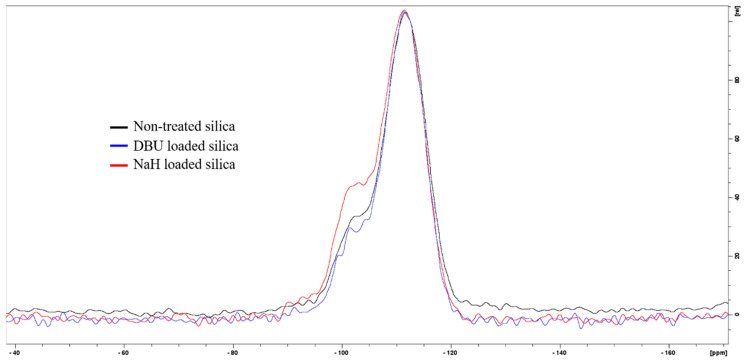
^29^Si ssNMR of untreated silica (black), DBU−loaded silica (blue), and NaH−loaded silica (red).

**Figure 3 polymers-14-01676-f003:**
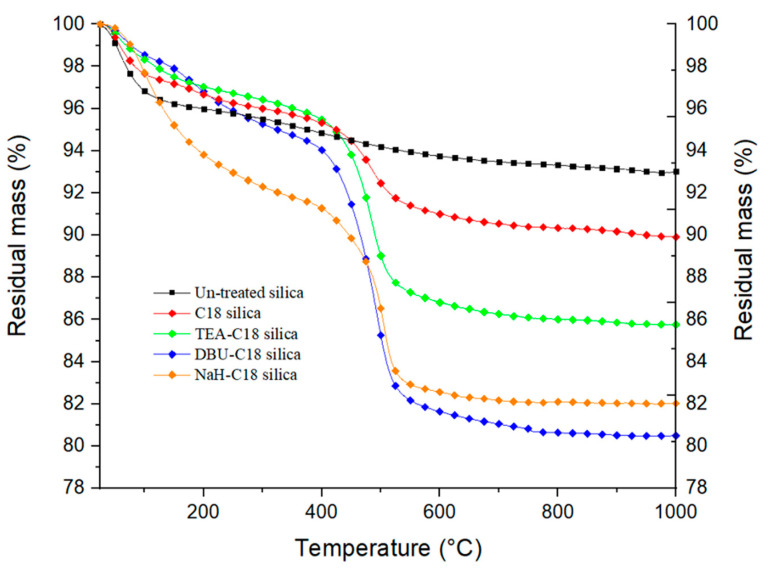
Thermogravimetric curves of untreated and C18−silanized silica.

**Figure 4 polymers-14-01676-f004:**
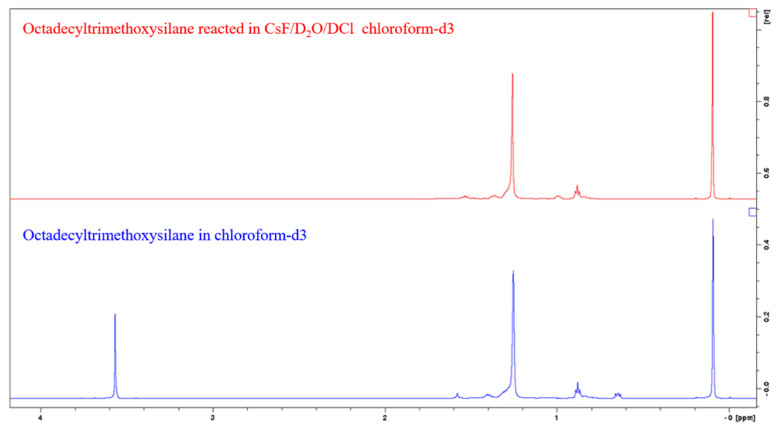
^1^H spectra of C18 silane in chloroform (**bottom**), and reacted in CsF/D_2_O/DCl and extracted with chloroform (**top**).

**Figure 5 polymers-14-01676-f005:**
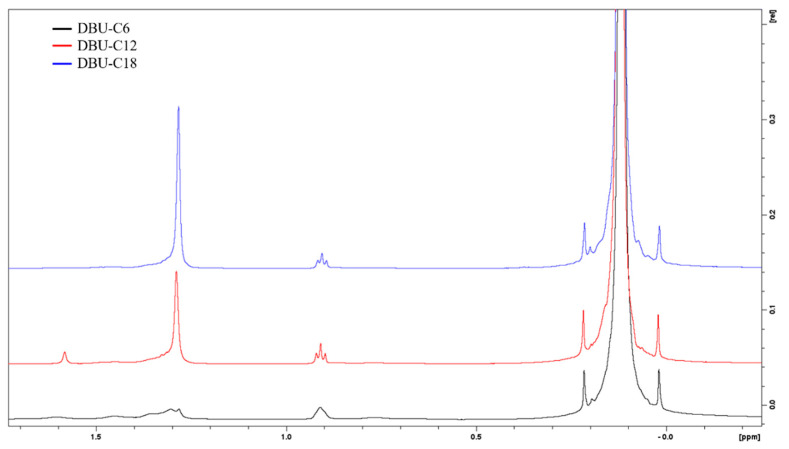
^1^H NMR spectra of C6 (**bottom**), C12 (**middle**), and C18 (**top**) silanized silica.

**Figure 6 polymers-14-01676-f006:**
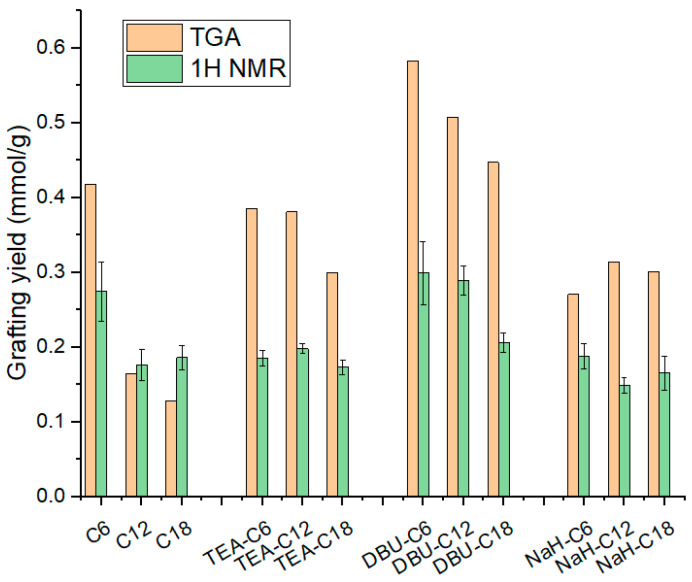
Silica silanization grafting yield from TGA and ^1^H NMR.

**Figure 7 polymers-14-01676-f007:**
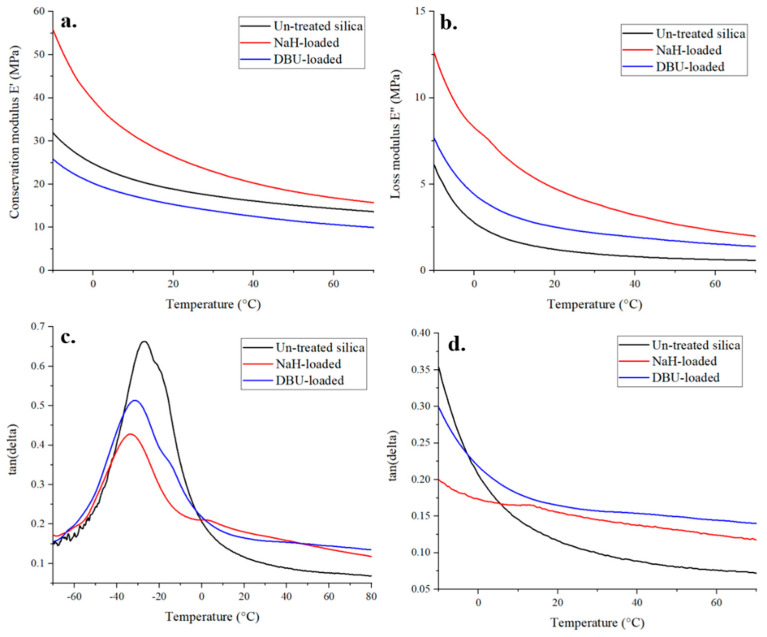
DMA curves of base−loaded silica−rubber composites—E′ (**a**), E″ (**b**) and tan(δ) (**c**,**d**).

**Table 1 polymers-14-01676-t001:** Composition and mixing steps for silica-rubber composite fabrication.

	Components	phr	Mixing Conditions
Step 1	Polystyrene-butadiene	80	80 °C for 10′
	Polybutadiene	20	
	TDAE Oil	25	
	Zinc oxide	0.5	
	Stearic acid	3	
	Silica	65	
Step 2	Step 1 compound	-	80 °C for 7′
	Silica	15	
	Silane (TESPD)	8	
	6PPD	2.5	
Step 3	Step 2 compound	-	60 °C for 1′45″
	Zinc oxide	2	
	Sulfur	1.1	
	MBT	0.3	
	DPG	3.2	
	CBS	2.3	

**Table 2 polymers-14-01676-t002:** Silica silanization grafting yield calculated from TGA and ^1^H NMR (in mmol/g).

	C6	C12	C18	TEA-C6	TEA-C12	TEA-C18	DBU-C6	DBU-C12	DBU-C18	NaH-C6	NaH-C12	NaH-C18
**TGA**	0.418	0.165	0.128	0.383	0.381	0.3	0.582	0.508	0.447	0.271	0.314	0.301
**^1^H NMR**	0.274	0.176	0.186	0.185	0.198	0.173	0.299	0.289	0.206	0.188	0.149	0.165

## Data Availability

Data is contained within the article or Appendix A.

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
