# Peer review of "Original Basic Activation for Enhancing Silica Particle Reactivity: Characterization by Liquid Phase Silanization and Silica-Rubber Nanocomposite Properties"

_polymers, 2022, doi:10.3390/polym14091676_

Round 1

Reviewer 1 Report

Silica particles have been considered efficient fillers in many composite applications. Optimal reaction modification methods in composite materials will improve their efficiency. Therefore, the subject of this research can be interesting for readers. However, a few tips are recommended to improve the quality of the manuscript.

  • The article's sentence structure and writing style should be more transparent. Some used statements make it difficult to understand topics. It is recommended to consider the fluency of the writing style for the whole text when reviewing the manuscript.
  • To better understand the readers, it is necessary to clarify the need for research in this manuscript. It is also better to enrich the research literature by introducing practical and tested cases of previous researchers' experiments.
  • The manuscript has well investigated the reactivity of silica particles in the basic state and the enhanced method. However, the quality of images and graphic evidence is not good. It is recommended to improve the quality of the figures used.

Reviewer 2 Report

The main question arising after reading is the doubt in the nature of the base acting in the reaction with NaH. The authors mention that gas evolves during interaction of NaH with silica. It makes to think about NaH hydrolysis induced by moisture from silica. In this case the authors estimate NaOH rather tnan NaH action. I would advice to take it into consideration and to refute this possibility in the text if this is not the case.

There are also several minor corrections required. The unit "phr" in the Table 1 is exotic. It needs wider explanation for the prospective readers, who don't deal with rubber production. Is it mass fraction? In Fig. 4 the letter o is missed in the name of the alcoxysilane, in both cases.
